# Method for Automatic Estimation of Instantaneous Frequency and Group Delay in Time–Frequency Distributions with Application in EEG Seizure Signals Analysis

**DOI:** 10.3390/s23104680

**Published:** 2023-05-11

**Authors:** Vedran Jurdana, Miroslav Vrankic, Nikola Lopac, Guruprasad Madhale Jadav

**Affiliations:** 1Faculty of Engineering, University of Rijeka, 51000 Rijeka, Croatia; jguruprasad@riteh.hr; 2Faculty of Maritime Studies, University of Rijeka, 51000 Rijeka, Croatia; nikola.lopac@pfri.uniri.hr; 3Center for Artificial Intelligence and Cybersecurity, University of Rijeka, 51000 Rijeka, Croatia

**Keywords:** time–frequency distributions, Rényi entropy, instantaneous frequency, group delay, EEG

## Abstract

Instantaneous frequency (IF) is commonly used in the analysis of electroencephalogram (EEG) signals to detect oscillatory-type seizures. However, IF cannot be used to analyze seizures that appear as spikes. In this paper, we present a novel method for the automatic estimation of IF and group delay (GD) in order to detect seizures with both spike and oscillatory characteristics. Unlike previous methods that use IF alone, the proposed method utilizes information obtained from localized Rényi entropies (LREs) to generate a binary map that automatically identifies regions requiring a different estimation strategy. The method combines IF estimation algorithms for multicomponent signals with time and frequency support information to improve signal ridge estimation in the time–frequency distribution (TFD). Our experimental results indicate the superiority of the proposed combined IF and GD estimation approach over the IF estimation alone, without requiring any prior knowledge about the input signal. The LRE-based mean squared error and mean absolute error metrics showed improvements of up to 95.70% and 86.79%, respectively, for synthetic signals and up to 46.45% and 36.61% for real-life EEG seizure signals.

## 1. Introduction

Electroencephalogram (EEG) recordings are widely used for assessing brain disorders [1,2,3]. EEG is a noninvasive approach for detecting and predicting seizures [4,5,6,7], which can be difficult to identify in infants. Recurrent seizures are the hallmark of epilepsy, one of the most prevalent neurological disorders in humans. Identifying seizures in EEG recordings typically requires real-time observation by a neurologist, leading to growing interest in automated methods for seizure detection.

EEG signals are nonstationary, and therefore, time–frequency (TF) and time-scale representations are commonly used for their analysis [6,8,9,10,11,12,13,14,15,16,17,18,19,20,21]. A time–frequency distribution (TFD) enables us to describe signal energy simultaneously in time and frequency. However, the most widely used TFDs, the quadratic TFDs (QTFDs), create highly oscillatory artifacts, known as cross-terms, for signals with several components or at least one nonlinear frequency-modulated (non-LFM) component [22,23,24]. Although the 2D low-pass filters in the ambiguity function (AF) domain are often used to suppress cross-terms, they may also suppress important components known as autoterms, resulting in a trade-off between cross-term reduction and autoterm resolution. To overcome this trade-off, a variety of filtering methods in the TF domain have been developed [22,25] as an alternative to conventional filtering methods utilizing nonstationary signals in the time domain [26,27,28]. One of these advanced filtering methods that outperforms others is the adaptive directional TFD (ADTFD), which achieves high resolution for multicomponent signals having multiple directions of energy distribution in the TF domain, such as EEG seizure signals [7,29,30,31,32].

Instantaneous frequency (IF) provides vital information on the time-varying spectral changes in nonstationary signals. In the TF signal analysis, a number of IF estimation methods have been developed [29,33,34,35,36,37,38,39,40,41,42,43,44]. However, IF is not as useful to characterize signals composed of spikes, where an infinite amount of frequencies is present [45,46,47]. Such behavior is present in EEG seizure signals, for which the group delay (GD) has been proven to be a better alternative to IF [46,47]. However, the limitation of the IF estimation method proposed in [48], and hence, the approach in [46,47], is that the number of components is required as input from a user, which limits its usage when no a priori information is available and when the number of components varies over time.

In this paper, we present a novel method for generating a binary map that automatically identifies distinct regions within the signal’s TFD, where either the IF or GD measure is particularly suitable for analysis. Our method utilizes information about the local number of signal components obtained from the localized Rényi entropy (LRE) information, namely the short-term Rényi entropy (STRE) [33,34] and the narrow-band Rényi entropy (NBRE) [35,36], to detect components and segment the TFD accordingly. Unlike other methods that require a priori knowledge of the number of components, our approach is applicable to signals with unknown and time-varying numbers of components.

The resulting binary map is used in a joint IF and GD estimation method, which was applied to two commonly used IF estimation algorithms in EEG signal analysis, namely image-based [22,37,38] and blind-source separation (BSS) [15,39,40]. These methods were selected for their suitability in an automatic environment, as they effectively use the component time support information from the STRE. Our approach further incorporates the frequency support information from the NBRE to estimate the GD and reduce the dependency of the IF algorithms on the STRE accuracy, which can be achieved for the considered signal examples.

Furthermore, we demonstrate that the shrinkage operator proposed in [35,36] for sparse TFD reconstruction can be effectively used for IF and GD estimation, with competitive performance compare with the image-based and BSS algorithms. We evaluate the performance of our method using mean squared error (MSE) between the local number of signal components before and after estimation, and we show the superior performance of our combined IF and GD estimation approach over IF estimation alone on both synthetic signals with additive noise and real-life EEG seizure signals.

The rest of this paper is organized as follows. Background theory, the proposed method and the EEG dataset used in this study are described in Section 2. The obtained results are thoroughly presented and discussed in Section 3. Finally, the paper’s conclusions are summarized in Section 4.

## 2. Materials and Methods

### 2.1. Time–Frequency Signal Analysis

A multicomponent nonstationary signal, denoted as z(t), is defined as the analytic associate of a real signal s(t) as
(1)$$z\left(t\right)=\sum_{i=1}^{NC}a_{i}\left(t\right)e^{j\varphi_{i}\left(t\right)},$$
where NC is the number of components, while $a_{i}\left(t\right)$ and $\varphi_{i}\left(t\right)$ denote the instantaneous amplitude and instantaneous phase of the signals *i*-th component, respectively. The ideal TFD, $\hat{\rho }\left(t,f\right)$, is a unit delta function following the crests of the ridges which represent the IF, $f_{0_{i}}\left(t\right)$, of the *i*-th component: (2)$$\hat{\rho }\left(t,f\right)=\sum_{i=1}^{NC}a_{i}^{2}\left(t\right)\delta \left(f-f_{0_{i}}\left(t\right)\right),$$
(3)$$f_{0_{i}}\left(t\right)=\frac{1}{2\pi }\frac{d}{dt}\arg z\left(t\right)=\frac{1}{2\pi }\frac{d\varphi_{i}\left(t\right)}{dt},$$
indicating the dominant frequency of the signal’s *i*-th component at a given time. A dual (or inverse) of the IF, namely the GD, $\tau_{d_{i}}\left(f\right)$, indicates the dominant time of the signal’s *i*-th component at a given frequency: (4)$$\tau_{d_{i}}\left(f\right)=-\frac{1}{2\pi }\frac{d}{df}\arg Z\left(f\right),$$
where Z(f) is the Fourier transform of z(t). The definitions of IF and GD are closely related, involving interchanging the time and frequency variables, with an extra minus sign in Equation (Equation 4). In practice, the IF is obtained by determining the signal’s component ridge across time slices of its TFD, while the GD is determined by determining the signal’s component ridge across frequency slices of the TFD. In the vast majority of instances, the ideal TFD is not achievable, because practical TFDs are not precisely localized and may be affected by cross-terms [22].

The Wigner–Ville Distribution (WVD) is widely used as the most fundamental TFD, defined as [22]: (5)$$W\left(t,f\right)=\int_{-\infty }^{\infty }z\left(t+\frac{\tau }{2}\right)z^{*}\left(t-\frac{\tau }{2}\right)e^{-j2\pi f\tau }d\tau ,$$
and it provides an estimate of the IF/GD for a signal with a single LFM component in the TF plane that is almost perfect. Yet, the cross-term vulnerability (when dealing with multicomponent signals) necessitates proper cross-terms suppression. Using the AF, A(ν,τ), calculated as
(6)$$A\left(\nu ,\tau \right)=\int_{-\infty }^{\infty }\int_{-\infty }^{\infty }W_{z}\left(t,f\right)e^{j2\pi (f\tau -\nu t)}dtdf,$$
the highly oscillatory cross-terms can be suppressed with a 2D low-pass filter, defining a QTFD class of TFDs, ρ(t,f): (7)A(ν,τ)=A(ν,τ)g(ν,τ),
(8)$$\rho \left(t,f\right)=\int_{-\infty }^{\infty }\int_{-\infty }^{\infty }A\left(\nu ,\tau \right)e^{j2\pi (\nu t-f\tau )}d\nu d\tau ,$$
where $g\left(\nu ,\tau \right)$ is the low-pass filter kernel in the AF. The conventional approaches to kernel design typically entail a compromise between the concentration of autoterms and the suppression of cross-terms [22].

### 2.2. Adaptive Directional TFD

To circumvent the above problem of the conventional TFDs, the adaptive directional TFD (ADTFD) that adjusts the direction of the smoothing kernel at each TF point in the TF plane is introduced [7,29], and it is mathematically expressed as follows:(9)$$\rho_{\left(ad\right)}\left(t,f\right)=\rho \left(t,f\right)\underset{t}{*}\underset{f}{*}\;\gamma_{\theta }\left(t,f\right),$$
where $\gamma_{\theta }\left(t,f\right)$ is the smoothing kernel whose direction is controlled by θ, while the double asterisk denotes double convolution in *t* and *f*. In this work, we used the Extended Modified B Distribution (EMBD) as an underlying QTFD with its kernel:(10)$$g\left(\nu ,\tau \right)=\frac{\int_{-\infty }^{\infty }\cosh^{-2\beta_{E}}\left(t\right)e^{j2\pi \nu t}dt}{\int_{-\infty }^{\infty }\cosh^{-2\beta_{E}}\left(t\right)dt}\cosh^{-2\alpha_{E}}\left(\tau \right),$$
where $\alpha_{E}=\beta_{E}=0.25$ are the time and frequency smoothing parameters [7,29,31,32]. As $\gamma_{\theta }\left(t,f\right)$, we selected the double-derivative directional Gaussian filter (DGF) as in [7,29,31,32]:(11)$$\gamma_{\theta }\left(t,f\right)=\frac{ab}{2\pi }\frac{d^{2}}{df_{\theta }^{2}}e^{-a^{2}t_{\theta }^{2}-b^{2}f_{\theta }^{2}},$$
where $t_{\theta }=t\cos\left(\theta \right)+f\sin\left(\theta \right)$ and $f_{\theta }=-t\sin\left(\theta \right)+f\cos\left(\theta \right)$, while parameters *a* and *b* control the extent of smoothing along the time and frequency axes. The DGF has low-pass characteristics along the time axis $\left(e^{-a^{2}t_{\theta }^{2}}\right)$, while it performs second-order differentiation along the frequency axis $\frac{ab}{2\pi }\left(\frac{d^{2}}{df_{\theta }^{2}}e^{-b^{2}f_{\theta }^{2}}\right)$. The direction angle of $\gamma_{\theta }\left(t,f\right)$ is adapted locally for each point in the TF domain by maximizing the correlation between the $\gamma_{\theta }\left(t,f\right)$ and TF ridges as
(12)$$\theta \left(t,f\right)=\arg\max_{\theta }|\left|\rho \left(t,f\right)\right|\underset{t}{*}\underset{f}{*}\;\gamma_{\theta }\left(t,f\right)|,$$
where −π/2≤θ≤π/2. The implementation of directional smoothing results in the suppression of cross-terms and the enhancement of autoterms. The optimization of smoothing kernel parameters and shape is necessary for achieving optimal performance, as they are dependent on the signal being analyzed. Previous studies [7,29,31,32] have indicated that assigning a small value to parameter *a* results in intensive smoothing along the major axis, whereas a larger value for parameter *b* prevents the merging of close components. To be more precise, a∈[2,3], while b∈[5,30]. In addition to shape parameters *a* and *b*, the window length, WL, of the $\gamma_{\theta }\left(t,f\right)$ affects the performance of the ADTFD. A filter with a small WL value fails to resolve close components and eliminate cross-terms, but it preserves the energy of short-duration components. Conversely, a larger WL value achieves the opposite effect. The computational demand of the exhaustive search involving all possible combinations of (a,b,WL) is significant. Therefore, we used a method for the automatic parameter optimization of ADTFD, namely the locally adaptive-ADTFD (LO-ADTFD) proposed in [31], where the final LO-ADTFD is obtained by choosing TF points with the minimum value from a given set of ADTFDs $\left\{\rho_{{\left(ad\right)}_{1}}\left(t,f\right),\rho_{{\left(ad\right)}_{2}}\left(t,f\right),\ldots \right\}$ and their respective parameters $\left\{\left(a_{1},b_{1},WL_{1}\right),\left(a_{2},b_{2},WL_{2}\right),\ldots \right\}$:(13)$$\rho_{\left(lo\right)}\left(t,f\right)=\min_{k}\left(\rho_{{\left(ad\right)}_{k}}\left(t,f\right)\right),$$
where $\rho_{{\left(ad\right)}_{k}}\left(t,f\right)$ is the *k*-th ADTFD in the defined set. That way, the LO-ADTFD preserves the energy of short-duration signals while achieving high-resolution TFD with resolved close components and suppressed cross-terms. In this work, we selected the parameter (a,b) values from the following set {(3,6),(3,8),(2,20),(2,30)}, while WL was optimized for each (a,b) pair using the concentration measure proposed in [41]:(14)$$M=\frac{1}{N_{t}N_{f}}{\left(\sum_{t=1}^{N_{t}}\sum_{f=1}^{N_{f}}{\left|\rho_{\left(ad\right)}\left(t,f\right)\right|}^{\frac{1}{2}}\right)}^{2},$$
where $N_{t}$ and $N_{f}$ denote the numbers of time samples and frequency bins, respectively, in range $\left[N_{t}/8:4:N_{t}/4\right]$ for the pairs {(3,6),(2,20)} and $\left[N_{t}/4:4:3N_{t}/8\right]$ for the pairs {(3,8),(2,30)}, as suggested in [31].

### 2.3. The Localized Rényi Entropy

The Rényi entropy, denoted by R(ρ(t,f)), is a comprehensive metric for signal complexity in the TF plane [42,43,44], defined as
(15)$$R\left(\rho \left(t,f\right)\right)=\frac{1}{1-\alpha_{R}}\log_{2}\int_{-\infty }^{\infty }\int_{-\infty }^{\infty }{\left(\frac{\rho (t,f)}{\int_{-\infty }^{\infty }\int_{-\infty }^{\infty }\rho \left(t,f\right)\;dtdf}\right)}^{\alpha_{R}}dtdf,$$
where for the odd integer parameter $\alpha_{R}>2$ the cross-terms get integrated out from the QTFD, ρ(t,f), which is normalized with respect to its total energy [34,42].

To refine the global approach mentioned earlier, the counting attribute of the Rényi entropy has been utilized to extract the local number of signal components from ρ(t,f) using STRE [34]. To achieve this, the Rényi entropy of the extracted signal’s TFD is compared with that of a reference TFD with a known number of component as follows: (16)$$NC_{t}^{\rho (t,f)}\left(t_{0}\right)=2^{R\left(\Gamma_{t_{0}}\left\{\rho \left(t,f\right)\right\}\right)-R\left(\Gamma_{t_{0}}\left\{\rho_{\mathrm{ref}}\left(t,f\right)\right\}\right)},$$
where *t* denotes localization using time slices, and $t_{0}$ is the observed time slice, while $\rho \left(t,f\right)$ and $\rho_{\mathrm{ref}}\left(t,f\right)$ denote the considered and reference TFD, respectively. The time-localization operator $\Gamma_{t_{0}}$ sets all TFD samples to zero, except those in the vicinity of $t_{0}$: (17)$$\Gamma_{t_{0}}\left\{\rho \left(t,f\right)\right\}=\left\{\begin{array}{cc} \rho (t,f), & t\in [t_{0}-\Theta_{t}/2,t_{0}+\Theta_{t}/2],\\ 0, & \mathrm{otherwise}, \end{array}\right.$$
where $\Theta_{t}$ is the parameter controlling the time-window length. The reference signal is a cosine signal with an amplitude of 1 and a constant normalized frequency of 0.1, providing a reference energy of a single component in each time slice [34].

In [35,36], more research reveals the weaknesses of STRE for specific signal types and introduces NBRE to counteract them. Using NBRE, one may determine the local number of signal components per frequency slice, $f_{0}$, by substituting the frequency-localization operator for the time-localization parameter in Equation (Equation 16) with the frequency-localization operator: (18)$$\Gamma_{f_{0}}\left\{\rho \left(t,f\right)\right\}=\left\{\begin{array}{cc} \rho (t,f), & f\in [f_{0}-\Theta_{f}/2,f_{0}+\Theta_{f}/2],\\ 0, & \mathrm{otherwise}, \end{array}\right.$$
where $\Theta_{f}$ is the frequency window length. The reference signal is a delta function centered at t=15 [36]. Note that $\rho \left(t,f\right)$ and $\rho_{\mathrm{ref}}\left(t,f\right)$ have to be obtained with the same TFD with reduced interference in order for the comparison to be valid.

The comparison between local numbers of components obtained from STRE and NBRE are shown in Figure 1 for the synthetic signal $z_{4\mathrm{LFM}}$, with $N_{t}=256$ samples composed of four LFM components with different amplitudes embedded in additive Gaussian noise (AWGN) with a signal-to-noise ratio (SNR) SNR=3 dB. Figure 1b,c show the reasoning behind introducing NBRE in [35,36]—an inaccurate increase in the local number of signal components, $NC_{t}\left(t\right)$, is evident for signals whose components are more aligned with the frequency axis (i.e., deviate from the method’s reference component), such as the signal $z_{4\mathrm{LFM}}$, whose LO-ADTFD is shown in Figure 1a. On the other hand, when signals have components more aligned with the time axis, STRE presents more accurate estimations of the local number of signal components [35,36].

### 2.4. Multicomponent Instantaneous Frequency Estimation Algorithms

#### 2.4.1. Image-Based Method

The first method for estimating the IF is an image processing method [37], which is divided into two sequential steps. In the first step, local peaks of the TFD are detected using its first and second derivatives as
(19)$$B^{\left(\mathrm{IM}\right)}\left(t,f\right)=\left\{\begin{array}{cc} 1, & \frac{d}{df}\rho \left(t,f\right)=0\;\;\mathrm{and}\;\;\frac{d^{2}}{df^{2}}\rho \left(t,f\right)<0,\\ 0, & \mathrm{otherwise}, \end{array}\right.$$
which generates a binary (t,f) image consisting of ones on all peak locations and zeros on all other points. This step usually provides peaks that do not belong exclusively to the autoterms. Hence, in the second step, the IFs of signal components are extracted by using the *m*-connectivity criterion derived from image processing. In this work, we used m=10, as in [7,22,37], which defines a 10-neighborhood set for a detected peak at location (x,y) as {(x−1,y),(x−1,y−1),(x−1,y+1),(x−1,y+2),(x−1,y−2),(x+1,y),(x+1,y−1),(x+1,y+1),(x+1,y+2),(x+1,y−2)}. According to this criterion, the points above and below the IF curves are not included, meaning that there can be only one frequency at any time instant for any given signal component.

Finally, a threshold must be specified for the minimum time duration of a valid signal component so that the final TFD contains only components that meet the threshold. In our example, where no prior knowledge of the input signal is available, the threshold is determined by the least component time support information provided by the STRE. This method demonstrated good computing efficiency and performance for real-world signals without requiring previous knowledge of the components’ IF laws and amplitudes [7,22,37,38].

#### 2.4.2. Blind-Source Separation Method

The second IF estimation method applied in this study is the blind-source separation (BSS) method, which is an efficient method for the localization and extraction of components from multicomponent signals in the TF domain [39]. The term “blind” refers to the provision of a mixture of statistically independent components without prior knowledge of its structure or number of components. The STRE’s information on the time supports of components is included in the version of the approach utilized in this study, namely the BSS-STRE [40], thus removing the need for several thresholds required by the original method [39]. The steps of the BSS component extraction method are summarized below.

First, the TFD of a signal is computed, ρ(t,f), and the corresponding $NC_{t}\left(t\right)$ from the STRE is obtained. The algorithm then locates the largest TFD peak at $\left(t_{0},f_{0}\right)$ and calculates the adaptive neighboring component frequency band, $\Delta B=B_{L}+B_{R}$, for the time slice $\rho (t_{0},f_{0}-B_{L}:f_{0}+B_{R})$. Next, the component is extracted at time-slice $t_{0}$, and $NC_{t}\left(t_{0}\right)$ is reduced by 1. Following that, the previous steps of the method are repeated in both directions around $t_{0}$ as $t_{0}\leftarrow t_{0}-1$ and $t_{0}\leftarrow t_{0}+1$ until the component edges are detected by the first derivation of $|NC_{t}\left(t\right)|$, $|NC_{t}^{\prime }\left(t\right)|\neq 0$. If there is at least one component remaining in ρ(t,f), the above steps are repeated for the succeeding component. The full pseudocode of this method may be found in [15,40].

The BSS method results in several TFDs, each containing a single extracted component from which the component IF is estimated in a separate vector. In order to be comparable with the $B^{\left(\mathrm{IM}\right)}\left(t,f\right)$, all estimated IFs will be displayed within a single binary TFD, denoted by $B^{\left(\mathrm{BSS}\right)}\left(t,f\right)$.

Note that the ADTFD, or more precisely, its automatically optimized version LO-ADTFD, is used as the underlying TFD in both estimation methods for this work, as its superior performance has been demonstrated in numerous IF estimation applications studies [38,45,46].

#### 2.4.3. Limitations of the Considered IF Estimation Algorithms Based on STRE

The accuracy of IF estimation methods is highly dependent on the accuracy of STRE. The findings in [35,36] indicate that STRE is unsuitable for signals whose components deviate from the time axis, resulting in artificially increased $NC_{t}\left(t\right)$ and reduced estimation accuracy. Since this issue may occur in real-life signals, the question arises as to which IF estimation approach should be used for such signals. An incorrect $NC_{t}\left(t\right)$ in the image-based IF estimation method can lead to either too low a threshold, resulting in interference being classified as a true signal component, or too high a threshold, causing some true signal components to be rejected. In the case of the BSS method, an incorrect $NC_{t}\left(t\right)$ can result in incomplete extraction of signal components, and higher $NC_{t}\left(t\right)$ values may cause the estimation of interference IFs.

In this paper, we aim to demonstrate the negative consequences of using an inappropriate localization approach in estimating the IFs of signal components. Specifically, we show that such an approach can cause estimated IFs to be discontinuous and shifted away from the true component ridge. Although polynomial functions can approximate discontinuous estimation samples, the approximation error increases as more estimated samples do not belong to the autoterms. This phenomenon is particularly problematic for the image-based method, which requires a larger *m*-connectivity criterion to connect true signal components. Otherwise, discontinuous signal components are often classified as interference and fail to meet the threshold criteria. Increasing the *m*-connectivity criterion is not recommended, as it can lead to the linking of interference terms. Our paper aims to address these limitations by proposing the use of the frequency localization approach and the estimation of the GD with information from the NBRE method for certain signals. By doing so, we can minimize the negative effects of an inappropriate localization approach on IF estimation.

### 2.5. The Proposed Rényi-Entropy-Based Method for Component Alignment Detection towards Time or Frequency Axis

In this section, we present a novel method for the automatic detection of TFD regions that require a time or frequency localization strategy. Such a strategy involves using TFD analysis with time or frequency slices, linked to an estimate of IFs or GDs. To detect such regions, we implemented the STRE and NBRE methods, which are sensitive to estimate errors when dealing with signal components that vary from the methods’ respective reference components. In this method, this sensitivity is turned into an advantage, as a considerable increase in the local number of signal components can indicate the need for an alternate localization strategy in a certain TFD region.

Next, the proposed method validates the detected increase in the local number of signal components by comparing the quality of IF and GD estimations in the identified TFD region. We observed that an incorrect localization approach can lead to discontinuous and inaccurate IF or GD estimations.

To estimate the IF and GD trajectories, we employed an approach based on the shrinkage operator proposed in [35,36] for sparse TFD reconstruction. The shrinkage operator, denoted with ${\mathrm{shrink}}_{t,f}$, operates independently for each time and frequency slice and removes samples that do not belong to the autoterms. The autoterms are locally associated with the $NC_{t}\left(t\right)$ or $NC_{f}\left(f\right)$ largest areas, where an area is calculated as a sum of samples between the minima to the left and right of the detected local maxima. This operator involves parameters $\delta_{t}$ and $\delta_{f}$, which control the number of samples around local maxima classified as autoterms [35,36]. Hence, by applying the shrinkage operator on desired TFD with parameters $\delta_{t}=\delta_{f}=1$, we may extract only local maxima belonging to signal autoterms, which basically represent the IF and GD estimations of signal components: (20)$$\rho^{t}\left(t,f\right)={\mathrm{shrink}}_{t}\left\{\rho \left(t,f\right)\right\}{|}_{\delta_{t}=1},\;\rho^{f}\left(t,f\right)={\mathrm{shrink}}_{f}\left\{\rho \left(t,f\right)\right\}{|}_{\delta_{f}=1},$$
where the t,f notation denotes shrinkage performed over time or frequency slices, while $\rho^{t}\left(t,f\right)$ and $\rho^{f}\left(t,f\right)$ denote the signal’s estimated IFs and GDs (or ridges), respectively.

To demonstrate the proposed method’s steps, we created a synthetic signal, $z_{\mathrm{mix}}\left(t\right)$, with $N_{t}=256$ samples, consisting of two constant FM and four non-LFM components with various different directions and time/frequency supports. Figure 2 illustrates the LO-ADTFD of the signal, as well as its estimated IFs, $\rho_{\left(lo\right)}^{t}\left(t,f\right)$, and GDs, $\rho_{\left(lo\right)}^{f}\left(t,f\right)$. Observe that the estimate quality of signal components is distinct between $\rho_{\left(lo\right)}^{t}\left(t,f\right)$ and $\rho_{\left(lo\right)}^{f}\left(t,f\right)$. Specifically, Figure 2b displays four discontinuous non-LFM components in $\rho_{\left(lo\right)}^{t}\left(t,f\right)$ whose alignment deviates from the time axis. Alternatively, Figure 2c depicts the same occurrence for $\rho_{\left(lo\right)}^{f}\left(t,f\right)$ but with inverse results. Now, the identical four non-LFM components exhibit stronger connectivity than two constant FM components that are aligned with the time axis.

To evaluate the quality of estimated IFs and GDs, we propose a metric based on the number of continuously connected regions of TFD samples denoted by $N_{r}$. Specifically, a TFD sample at location (x,y) is considered to be part of a region if it is connected to at least one sample in its 8-neighborhood set: {(x−1,y−1),(x−1,y),(x−1,y+1),(x,y−1),(x,y+1), (x+1,y−1),(x+1,y),(x+1,y+1)}. The resulting components are then counted to obtain $N_{r}$. This way, the proposed metric detects and penalizes discontinuities in the estimated IF or GD trajectories. Higher values of $N_{r}$ indicate lower consistency among the estimated components (i.e., autoterms), reflecting a lower quality of the estimated IF or GD trajectories. In our proposed method, the result of the $N_{r}$ metric serves as a decision-making factor in each TFD region under consideration.

The objective of our proposed method is to generate a binary component alignment map, denoted by BM(t,f), that distinguishes TFD regions with components suitable for time or frequency localization using ones and zeros, respectively. We provide a comprehensive description of the proposed method’s steps below:The first step in the proposed approach involves calculating the TFD for a given signal in the time domain, followed by estimation of the IFs and GDs using the shrinkage operator, resulting in $\rho^{t}\left(t,f\right)$ and $\rho^{f}\left(t,f\right)$, as shown in Figure 2.Now, the values of $N_{r}\left(\rho^{t}\left(t,f\right)\right)$ and $N_{r}\left(\rho^{f}\left(t,f\right)\right)$ are computed and compared. If $N_{r}\left(\rho^{t}\left(t,f\right)\right)\leq N_{r}\left(\rho^{f}\left(t,f\right)\right)$, the proposed method assumes that signal components are primarily aligned with the time axis and generates BM(t,f) using ones. Additionally, the STRE method is employed to calculate $NC_{t}\left(t\right)$, which is used to investigate local component behavior. Conversely, if $N_{r}\left(\rho^{t}\left(t,f\right)\right)>N_{r}\left(\rho^{f}\left(t,f\right)\right)$, the proposed method generates BM(t,f) using zeros and uses the NBRE with $NC_{f}\left(f\right)$ as the initial localization approach.The proposed algorithm examines the input $NC_{t}\left(t\right)$ or $NC_{f}\left(f\right)$ for pronounced local maxima, which may indicate an inadequate local component for the current STRE or NBRE approach. To identify such maxima, we first locate all local maxima within $NC_{t}\left(t\right)$ (or $NC_{f}\left(f\right)$), followed by the calculation of the difference in the local number of signal components, denoted as ΔNC, between the observed maximum and the minima to the left and right. We consider all ΔNC≥1.50 as “suspicious” intervals that require further analysis. The chosen threshold value of 1.50 is based on the desire to detect components that deviate from the respective time or frequency axis more than the LFM component, with a starting and stopping normalized frequency at 0 and 0.5, respectively. This component showed marginal accuracy for both approaches, with mean numbers of local components obtained as $\frac{1}{N_{t}}\sum_{t=1}^{N_{t}}NC_{t}\left(t\right)≊\frac{1}{N_{f}}\sum_{f=1}^{N_{f}}NC_{f}\left(f\right)≊1.48$. If all ΔNC values are less than 1.50, the algorithm outputs a BM(t,f) consisting of only ones or zeros and terminates. Otherwise, the algorithm proceeds to the next step for further analysis.Next, the algorithm identifies a segment of time (or frequency) slices from $NC_{t}\left(t\right)$ (or $NC_{f}\left(f\right)$), where the edges are defined by local minima satisfying ΔNC≥1.50. An example of $NC_{f}\left(f\right)$ for the signal $z_{\mathrm{mix}}\left(t\right)$ is shown in Figure 3a, where the first segment is indicated by red dashed lines at frequency bins $f_{1}$ and $f_{2}$. The same segment is then extracted from a TFD, within which an unsuitable signal component for the current localization approach may be present. Figure 3b illustrates an example of $\rho_{\left(lo\right)}\left(t,f\right)$ with the segment constrained by the previously detected frequency bins $f_{1}$ and $f_{2}$, where a constant FM component needs to be further detected in the subsequent algorithm steps as unsuitable for the frequency localization approach.At this point, additional localization is performed within the segmented TFD by computing the LRE in the opposite direction of the previous step. In particular, $NC_{f}\left(f\right)$ is calculated if $N_{r}\left(\rho^{t}\left(t,f\right)\right)\leq N_{r}\left(\rho^{f}\left(t,f\right)\right)$, and $NC_{t}\left(t\right)$ is calculated if $N_{r}\left(\rho^{t}\left(t,f\right)\right)>N_{r}\left(\rho^{f}\left(t,f\right)\right)$. We follow the same procedure as in the previous two steps, identifying all local maxima and minima with ΔNC≥1.50. Then, we detect all segments with borders satisfying ΔNC≥1.50, which define a 2D TF region or block within $\rho^{t}\left(t,f\right)$ and $\rho^{t}\left(t,f\right)$ that are evaluated in reference to the $N_{r}$ value. If $N_{r}$ is lower for a TFD block within $\rho^{t}\left(t,f\right)$ than in $\rho^{f}\left(t,f\right)$, it implies that a time localization technique is more suited, and BM(t,f) in positions specified with a TFD block is set to 1; otherwise, it is set to 0. All the remaining estimated signal components that are saturated inside minima with ΔNC<1.50 belong to the current localization approach (which differs from the localization approach in the previous step), and the corresponding TFD block within the final BM(t,f) changes its values from 1↔0. Finally, the $N_{r}$ value is compared in $\rho^{t}\left(t,f\right)$ and $\rho^{f}\left(t,f\right)$ for time samples or frequency bins where no component was detected (i.e., $NC_{f}\left(f\right)=0$ or $NC_{t}\left(t\right)=0$), and the BM(t,f) is set to 0 or 1 based on the lower $N_{r}$ value.Figure 4 illustrates the procedure described in this section. Firstly, an LO-ADTFD segment bounded in the range $\left[f_{1},f_{2}\right]$ is extracted and subjected to an opposite LRE approach, such as STRE, to obtain the $NC_{t}\left(t\right)$, as shown in Figure 4a,b. The red dashed lines in Figure 4b denote the segment detected using the $NC_{t}\left(t\right)$, which defines a TFD block $\left(t_{1}:t_{2},f_{1}:f_{2}\right)$ for evaluation in $\rho_{\left(lo\right)}^{t}\left(t,f\right)$ and $\rho_{\left(lo\right)}^{f}\left(t,f\right)$, shown in Figure 4c,d, respectively. The estimation of GDs within TFD blocks in red dashed lines exhibits superior connectivity than IF estimation, leading to the $BM(t_{1}:t_{2},f_{1}:f_{2})$ to remain unchanged from the initial values of zero. Since the constant FM component detected inside the green dashed lines does not produce inaccuracies in the $NC_{t}\left(t\right)$, as shown in Figure 4b, the $BM(t_{3}:t_{4},f_{1}:f_{2})$ is changed to one.Steps 4 and 5 are repeated for the remaining detected segments in the $NC_{t}\left(t\right)$ (or $NC_{f}\left(f\right)$) input, i.e., until all ΔNC≥1.50 are examined.

Figure 5 illustrates the BM(t,f) obtained for the signal $z_{\mathrm{mix}}\left(t\right)$, demonstrating that the proposed method effectively labeled both constant FM components for the time localization approach.

### 2.6. Component Extraction Using the Component Alignment Map

The BM(t,f) map obtained provides the means to extract signal components in the TFD. Extraction of these components is not achieved individually but through two sets of components: one suitable for localization through time slices and the other suitable for localization through frequency bins. The extraction process is simplified by multiplying the BM(t,f) map with the TFD using the operator κ{·}, which we defined in two ways. To derive the signal components corresponding to the localized approach through time slices, the operator $\kappa_{t}\left\{\cdot \right\}$ is used:(21)$$\kappa_{t}\left\{\rho \left(t,f\right)\right\}=\left\{\begin{array}{cc} \rho (t,f), & BM(t,f)=1,\\ 0, & BM(t,f)=0, \end{array}\right.$$
where, by multiplying the BM(t,f) map with TFD, only the regions defined by the units in BM(t,f) are retained in TFD. Likewise, the operator $\kappa_{f}\left\{\cdot \right\}$ is employed to obtain the components corresponding to the localized approach through frequency bins:(22)$$\kappa_{f}\left\{\rho \left(t,f\right)\right\}=\left\{\begin{array}{cc} \rho (t,f), & BM(t,f)=0,\\ 0, & BM(t,f)=1, \end{array}\right.$$
by means of which only the TFD regions that are defined by zeros in BM(t,f) are kept in TFD.

Consequently, when max{BM(t,f)}=1 and min{BM(t,f)}=0, the input TFD can be split into two TFDs, $\kappa_{t}\left\{\rho \left(t,f\right)\right\}$ and $\kappa_{f}\left\{\rho \left(t,f\right)\right\}$, enabling the local number of signal components to be computed using the STRE and NBRE methods, respectively. Since both $\kappa_{t}\left\{\rho \left(t,f\right)\right\}$ and $\kappa_{f}\left\{\rho \left(t,f\right)\right\}$ are expected to contain components corresponding to the chosen localization approach, more precise estimates of the local number of signal components can be obtained compared with those obtained from the original TFD.

Figure 6 illustrates the results of applying the operators $\kappa_{t}$ and $\kappa_{f}$ to the signal $z_{\mathrm{mix}}\left(t\right)$. The obtained $\kappa_{t}\left\{\rho_{\left(lo\right)}\left(t,f\right)\right\}$ and $\kappa_{f}\left\{\rho_{\left(lo\right)}\left(t,f\right)\right\}$, shown in Figure 6a,b, respectively, contain components suitable for the chosen localization approach. This is further corroborated by the $NC_{t}\left(t\right)$ and $NC_{f}\left(f\right)$ estimates in $\kappa_{t}\left\{\rho_{\left(lo\right)}\left(t,f\right)\right\}$ and $\kappa_{f}\left\{\rho_{\left(lo\right)}\left(t,f\right)\right\}$, respectively, which lack significant inaccurate local maxima. Notably, the $NC_{t}\left(t\right)$ and $NC_{f}\left(f\right)$ estimates obtained from the split TFDs are considerably more precise than those obtained from the original TFD, as demonstrated in Figure 6c,d. However, it is important to emphasize that the estimates of the local number of signal components should be interpreted in conjunction with BM(t,f), since they do not represent estimates of the entire TFD but rather of the TFD regions specified within BM(t,f).

### 2.7. Method for an Automatic Estimation of IF and GD

Upon completion of the necessary prerequisites, we propose a new method that can automatically estimate both the IF and GD of signal components in a TFD. This method leverages the use of the binary map BM(t,f) to identify the TFD regions that require IF or GD estimation. Meanwhile, the behavior of the signal components, or autoterms, is defined by the local number of signal components obtained through the STRE and NBRE methods applied on TFDs with extracted components, $\kappa_{t}\left\{\rho \left(t,f\right)\right\}$ and $\kappa_{f}\left\{\rho \left(t,f\right)\right\}$, respectively.

The proposed method is composed of the following steps:First, the input signal’s TFD is processed using the $\kappa_{t,f}\left\{\cdot \right\}$ operator to obtain two TFDs: one TFD, $\kappa_{t}\left\{\rho \left(t,f\right)\right\}$, containing signal components suitable for IF estimation using a time-slice approach, and another TFD, $\kappa_{f}\left\{\rho \left(t,f\right)\right\}$, containing signal components suitable for GD estimation using a frequency-slice approach.Next, the IFs are estimated from $\kappa_{t}\left\{\rho \left(t,f\right)\right\}$ using any IF estimation algorithm, the result of which is denoted as $B_{t}\left(t,f\right)$.A matrix transpose is applied to the discrete version of $\kappa_{f}\left\{\rho \left(t,f\right)\right\}$, which interchanges the time and frequency axes and enables GD estimation in frequency slices using the same IF estimation algorithm approach, resulting in $B_{f}\left(t,f\right)$.Finally, the IF and GD estimations are combined within a resulting binary TFD, B(t,f), which is an output of the proposed method summing $B_{t}\left(t,f\right)$ and $B_{f}\left(t,f\right)$.

The proposed method’s steps are visualized in Figure 7. We implemented the above-proposed method in the considered IF estimation methods, which may be now considered as IF/GD estimation algorithms utilizing both LRE methods, namely the image-based STRE-NBRE and BSS-STRE-NBRE.

It is worth noting that the combined estimates of the IF and GD may be extracted from the estimates generated by the operator ${\mathrm{shrink}}_{t,f}\left\{\cdot \right\}$, previously used in the proposed method for BM(t,f) as
(23)$$B^{\left(\mathrm{shrink}\right)}\left(t,f\right)=\kappa_{t}\left\{\rho^{t}\left(t,f\right)\right\}+\kappa_{f}\left\{\rho^{f}\left(t,f\right)\right\}.$$

In the following section, the performance of the $B^{\left(\mathrm{shrink}\right)}\left(t,f\right)$ estimation is compared with that of the image-based and BSS methods.

### 2.8. EEG Dataset Description

Seizure signals in EEG recordings are often modeled as multicomponent piecewise FM signals:(24)$$s\left(t\right)=\sum_{i=1}^{NC}a_{i}\left(t\right)e^{j2\pi \int f_{0_{i}}\left(\tau \right)d\tau }.$$However, this model does not take into consideration the spikes or short-duration transients that are regularly seen in EEG readings. In order to account for these spikes, the updated signal model that we use in this study is mathematically given as follows [7,30]:(25)$$s\left(t\right)=\sum_{i=1}^{NC}a_{i}\left(t\right)e^{j2\pi \int f_{0_{i}}\left(\tau \right)d\tau }+\sum_{i=1}^{NC}\delta \left(t-T_{i}\right),$$
where $T_{i}$ is a time-varying shift.

Analyzing signals that contain both rhythmic and spike features using traditional TF techniques is challenging, as they have energy distributed along both the time and frequency axes. Smoothing along the frequency axis can eliminate cross-terms created by spikes, but it also reduces the resolution of the sinusoidal signal components. In addition, such signals require the use of a combined IF and GD estimation method, as estimating only the IFs fails to recover spike features [47,48].

We used a database of 200 EEG seizure segments, which were previously uploaded as supplementary material in [38] and have been used in [7,30,32,47], from which an illustrative example was chosen, as used in additional studies [22,31]. The data and relevant code are publicly available at https://github.com/nabeelalikhan1/EEG-Classification-IF-and-GD-features (accessed on 1 October 2022). The EEG recordings were obtained from newborns at the NICU of the Royal Brisbane and Women’s Hospital, Brisbane, Australia, using the MEDELEC Profile System. Twelve electrodes were placed according to the international 10–20 standard, which were used to construct a 20-channel bipolar montage. The recordings underwent prefiltering using an analog bandpass filter with a bandwidth of 0.5 to 70 Hz. The signal was then sampled to 256 Hz before being digitally resampled to 32 Hz, as the majority of the signal energy is typically found below 12 Hz. The resulting signal segment is 8 s in duration and acquired at a sampling rate of 32 Hz, resulting in a total of $N_{t}=256$ samples [7,30,32,38,47]. Previous studies have shown that a differentiator filter can be used to whiten the EEG background and enhance the signature of spikes in EEG signals [30,47,49,50]. Hence, the proposed IF/GD estimation method’s performance was tested on EEG seizure signal without, denoted as $z_{\mathrm{EEG}}\left(t\right)$, and with a differentiatior filter, denoted as $z_{{\mathrm{EEG}}_{\mathrm{filt}}}\left(t\right)$.

## 3. Results and Discussion

We compared the performance of the combined IF and GD estimation approach with that of the IF estimation approach alone for both synthetic signals, $z_{\mathrm{LFM}}\left(t\right)$ and $z_{\mathrm{mix}}\left(t\right)$, and real-life EEG seizure signals, $z_{\mathrm{EEG}}\left(t\right)$ and $z_{{\mathrm{EEG}}_{\mathrm{filt}}}\left(t\right)$. It is important to note that for IF estimation only, algorithms use the STRE method applied to an input LO-ADTFD. However, for IF and GD estimation, the algorithms utilize the proposed method with BM(t,f), along with the STRE and NBRE methods applied to the extracted components from an input LO-ADTFD using the proposed operators $\kappa_{t}$ and $\kappa_{f}$. We calculated the STRE and NBRE using the parameter $\alpha_{R}=3$, with $\Theta_{t}=\Theta_{f}=11$ for the synthetic signals $z_{\mathrm{LFM}}\left(t\right)$ and $z_{\mathrm{mix}}\left(t\right)$ and $\Theta_{t}=\Theta_{f}=5$ for the signals $z_{\mathrm{EEG}}\left(t\right)$ and $z_{{\mathrm{EEG}}_{\mathrm{filt}}}\left(t\right)$, in order to capture spike features more precisely, which have been shown to be stable in [33,35,36,51].

To supplement the visual inspection of the results, it is important to quantify the impact of missing estimated IFs and GDs on the connectivity of components in the signal. Since the loss of components can occur at any point in the TFD, it is necessary to use a performance indicator that can monitor local components. To achieve this, we utilized an LRE-based indicator that has been shown to be effective in detecting reconstructed TFDs with discontinuous autoterms in prior work [36]. Specifically, we employed two performance indicators that measure the error between the local number of signal components in the original LO-ADTFD with fully preserved auto terms ($\rho_{\left(lo\right)}\left(t,f\right)$) and the TFD with estimated IFs/GDs (B(t,f)), using mean squared error (MSE) given as
(26)$${\mathrm{MSE}}_{t}=\frac{1}{N_{t}}\sum_{t=1}^{N_{t}}{\left(\frac{NC_{t}^{\rho_{\left(lo\right)}\left(t,f\right)}\left(t\right)-NC_{t}^{B(t,f)}\left(t\right)}{\max\left(NC_{t}^{\rho_{\left(lo\right)}\left(t,f\right)}\left(t\right),NC_{t}^{B(t,f)}\left(t\right)\right)}\right)}^{2},$$
(27)$${\mathrm{MSE}}_{f}=\frac{1}{N_{f}}\sum_{f=1}^{N_{f}}{\left(\frac{NC_{f}^{\rho_{\left(lo\right)}\left(t,f\right)}\left(f\right)-NC_{f}^{B(t,f)}\left(f\right)}{\max\left(NC_{f}^{\rho_{\left(lo\right)}\left(t,f\right)}\left(f\right),NC_{f}^{B(t,f)}\left(f\right)\right)}\right)}^{2},$$
(28)$${\mathrm{MSE}}_{t,f}=\frac{{\mathrm{MSE}}_{t}+{\mathrm{MSE}}_{f}}{2},$$
and mean absolute error (MAE) given as
(29)$${\mathrm{MAE}}_{t}=\frac{1}{N_{t}}\sum_{t=1}^{N_{t}}|\frac{NC_{t}^{\rho_{\left(lo\right)}\left(t,f\right)}\left(t\right)-NC_{t}^{B(t,f)}\left(t\right)}{\max\left(NC_{t}^{\rho_{\left(lo\right)}\left(t,f\right)}\left(t\right),NC_{t}^{B(t,f)}\left(t\right)\right)}|,$$
(30)$${\mathrm{MAE}}_{f}=\frac{1}{N_{f}}\sum_{f=1}^{N_{f}}|\frac{NC_{f}^{\rho_{\left(lo\right)}\left(t,f\right)}\left(f\right)-NC_{f}^{B(t,f)}\left(f\right)}{\max\left(NC_{f}^{\rho_{\left(lo\right)}\left(t,f\right)}\left(f\right),NC_{f}^{B(t,f)}\left(f\right)\right)}|,$$
(31)$${\mathrm{MAE}}_{t,f}=\frac{{\mathrm{MAE}}_{t}+{\mathrm{MAE}}_{f}}{2}.$$

Higher values of ${\mathrm{MSE}}_{t,f}$ and ${\mathrm{MAE}}_{t,f}$ indicate a greater amount of missing estimated IFs and/or GDs in B(t,f), suggesting that an inappropriate estimation strategy has been employed. Normalizing the local component count enabled a fair MSE and MAE comparison across signals.

### 3.1. Results for Synthetic Signals

The efficacy of the proposed BM(t,f) was evaluated on the synthetic signal example $z_{\mathrm{LFM}}\left(t\right)$, with the results presented in Figure 8. Since all four LFM components of the signal $z_{\mathrm{LFM}}\left(t\right)$ deviate from the time axis, the estimated GDs, shown in Figure 8b, offer a better connection than the estimated IFs, shown in Figure 8a. Consequently, the proposed method generates the BM(t,f) map, correctly highlighting the TFD regions along the signal’s components for the frequency localization approach, as shown in Figure 8c,d.

In the case of the signal $z_{\mathrm{LFM}}\left(t\right)$, Figure 9a demonstrates that the image-based STRE method was incapable of linking TF peaks that deviate from the time axis, resulting in the rejection of all four LFM autoterms and the estimation of interference terms that were almost parallel to the time axis. However, when utilizing the proposed combined IF and GD estimation, the autoterms were fully estimated with high connectivity (as shown in Figure 9b). Nevertheless, some interference and noise terms were also estimated due to the obtained spike in $NC_{f}\left(f\right)$ estimation shown in Figure 1c, which provided a frequency support threshold that is too small. While the BSS-STRE method did not experience difficulties with missing components, the estimated IFs were highly discontinuous, and not all of them belonged to the autoterms, as shown in Figure 9c. Conversely, Figure 9d demonstrates that the combined IF and GD estimation resulted in significantly improved component connectivity, with nearly all samples belonging to the autoterms.

Similar results were obtained for the signal $z_{\mathrm{mix}}\left(t\right)$. The image-based STRE method was unable to connect non-LFM components (or parts of components) that deviated from the time axis, as illustrated in Figure 10a. Meanwhile, the BSS-STRE method produced IF estimates that were discontinuous and dislocated from the true component ridge, as seen in Figure 10c. However, the proposed combined IF and GD estimation significantly improved the performance of both methods, as shown in Figure 10b,d. The performance of the $B^{\left(\mathrm{shrink}\right)}\left(t,f\right)$ obtained by the shrinkage operator is depicted in Figure 11. For both signal examples, the estimated component ridges in the TFD showed superior performance compared with those obtained using the image-based method, while being very similar to the BSS method, demonstrating high component connectivity and belonging to the autoterms of the signal.

The results presented in Table 1 demonstrate a significant reduction in ${\mathrm{MSE}}_{t,f}$ and ${\mathrm{MAE}}_{t,f}$ when using the proposed method for combined IF and GD estimation with STRE and NBRE information. For the signals $z_{\mathrm{LFM}}\left(t\right)$ and $z_{\mathrm{mix}}\left(t\right)$, the image-based method’s estimation improved by 71.25% and 81.11% in terms of ${\mathrm{MSE}}_{t,f}$ and 50.09% and 62.70% in terms of ${\mathrm{MAE}}_{t,f}$, respectively. The BSS method’s estimation also improved by 92.95% and 83.17% in terms of ${\mathrm{MSE}}_{t,f}$ and 82.37% and 65.07% in terms of ${\mathrm{MAE}}_{t,f}$, respectively.

Moreover, the results demonstrate that the $B^{\left(\mathrm{shrink}\right)}\left(t,f\right)$ obtained using the shrinkage-operator-based method for IF and GD estimation outperformed the image-based STRE and BSS-STRE methods—${\mathrm{MSE}}_{t,f}$ and ${\mathrm{MAE}}_{t,f}$ values improved by up to 95.62% and 86.52%, respectively, for the signal $z_{\mathrm{LFM}}\left(t\right)$ and 90.01% and 72.79%, respectively, for the signal $z_{\mathrm{mix}}\left(t\right)$. Moreover, the results show that the obtained $B^{\left(\mathrm{shrink}\right)}\left(t,f\right)$ is competitive with the $B^{\left(\mathrm{BSS}\right)}\left(t,f\right)$ obtained using the BSS-STRE-NBRE algorithm, showing a reduction in ${\mathrm{MSE}}_{t,f}$ for $z_{\mathrm{LFM}}\left(t\right)$ by 3.20%, while the rest of the indicators are slightly in favor of the BSS-STRE-NBRE method (by up to 2.01%).

#### Sensitivity to Noise

In this study, the proposed shrinkage-operator-based, BSS-STRE-NBRE and image-based STRE-NBRE methods for estimating IF and GD were evaluated for their robustness to noise. Synthetic signals, including $z_{\mathrm{LFM}}\left(t\right)$ and $z_{\mathrm{mix}}\left(t\right)$, were embedded in AWGN with an SNR that varied between 0 and 10 dB in 1000 independent simulations. The F1 score metric, which combines precision and recall, was used to evaluate the algorithm’s performance, given as
(32)$$\mathrm{Precision}=\frac{\mathrm{TP}}{\mathrm{TP}+\mathrm{FP}},$$
(33)$$\mathrm{Recall}=\frac{\mathrm{TP}}{\mathrm{TP}+\mathrm{FN}},$$
(34)$$F1=2\cdot \frac{\mathrm{Precision}\cdot \mathrm{Recall}}{\mathrm{Precision}+\mathrm{Recall}},$$
where true positives (TP) and true negatives (TN) indicate the number of samples that were correctly estimated as a signal or noise/interference component, respectively, while false positives (FP) and false negatives (FN) refer to the number of noise/interference or signal samples that were incorrectly identified as signal or noise, respectively. Note that F1 values range from 0 to 1, with higher values indicating better performance.

To further validate the F1 score metric, the 2D MSE between the noise-free and noisy TFDs was calculated using the scaled and squared Frobenius norm as follows: (35)$$F-\mathrm{norm}=\frac{1}{N_{t}N_{f}}||B\left(t,f\right)-B^{\left(\mathrm{noise}\right)}{\left(t,f\right)||}_{F}^{2}=\frac{1}{N_{t}N_{f}}\sum_{t=1}^{N_{t}}\sum_{f=1}^{N_{f}}{\left|B\left(t,f\right)-B^{\left(\mathrm{noise}\right)}\left(t,f\right)\right|}^{2}.$$The F-norm value was defined as the squared norm of the difference between the two TFDs, divided by the total number of time–frequency bins, $N_{t}\times N_{f}$. A lower F-norm value indicates better performance.

We evaluated the accuracy of the local number of signal component estimates by computing the MSE between $NC_{t}\left(t\right)$ and $NC_{f}\left(f\right)$ for both noise-free and noisy signals using Equation (3). The results, depicted in Figure 12, show that the LRE methods produce stable estimates for SNR values above 1 dB. Moreover, the noise sensitivity of all IF/GD estimation algorithms considered is determined by the sensitivity of LRE methods, as indicated by the F1 and F-norm values shown in Figure 13 and Figure 14. Notably, the image-based STRE-NBRE method is more sensitive to noise than the shrinkage-operator-based and BSS-STRE-NBRE methods, as even minor inaccuracies in the estimated local number of signal components can result in poor threshold values, as seen in Figure 9b.

### 3.2. Results for Real-Life EEG Seizure Signals

Figure 15a presents the time-domain representation of the signal $z_{\mathrm{EEG}}\left(t\right)$, revealing several spikes. These spikes are effectively captured in the TF domain using LO-ADTFD, as demonstrated in Figure 15b,c for the original and filtered signals, $z_{\mathrm{EEG}}\left(t\right)$ and $z_{\mathrm{EEG}\mathrm{filt}}\left(t\right)$, respectively, which also reveal an additional single sinusoidal component. Notably, the differentiator filter significantly reduced the background noise and enhanced the desired spike components, leading to a cleaner signal, $z_{{\mathrm{EEG}}_{\mathrm{filt}}}\left(t\right)$.

Figure 16 demonstrates that the proposed BM(t,f) effectively identified and separated the spike and sinusoidal components in both signals, $z_{\mathrm{EEG}}\left(t\right)$ and $z_{{\mathrm{EEG}}_{\mathrm{filt}}}\left(t\right)$. As a result, Figure 17 displays the extracted sinusoidal and spike components in separate TFDs. The local number of components, $NC_{t}\left(t\right)$ and $NC_{f}\left(f\right)$, estimated from these TFDs, shows a significant reduction in inaccurate local maxima compared with those obtained from the input LO-ADTFD, as illustrated in Figure 18.

Figure 19 displays the estimated IFs and GDs of the existing approach in [47] for the signals $z_{\mathrm{EEG}}\left(t\right)$ and $z_{\mathrm{EEG}\mathrm{filt}}\left(t\right)$. The IFs are estimated from the signal in the time domain, while the GDs are obtained from the Fourier transform of a signal using the duality property, which transposes a signal in the TF domain [47]. The results indicate that the approach is effective in estimating the IFs and GDs for the filtered EEG signal $z_{\mathrm{EEG}\mathrm{filt}}\left(t\right)$ used in the original study [47]. However, the approach proved unsuitable for estimating the GDs of the unfiltered signal $z_{\mathrm{EEG}}\left(t\right)$ due to the presence of background noise and the inconsistent number of components over the TF domain, leading to inaccurate GD estimates, as illustrated in Figure 19d. Furthermore, it is worth noting that the user needs to provide the global number of components to obtain these estimates, which may present a practical limitation when acquiring an unknown signal.

Figure 20 and Figure 21 compare the performance of the IF estimation methods against the proposed mutual IFs and GDs estimations. As the spike and sinusoidal components are perpendicular to each other, their estimated IFs exhibit completely opposite performances. Specifically, all IF estimation methods successfully estimated the IFs of the sinusoidal component, while the IF estimates of the spike components were highly discontinuous and, especially for the unfiltered signal $z_{\mathrm{EEG}}\left(t\right)$, indistinguishable from the background samples. However, estimating the GDs for the spike components significantly improved their connectivity and overall preservation, as shown in Figure 20 and Figure 21, using all considered IF and GD estimation methods.

The numerical results of the mutual IF and GD estimations, presented on illustrative $z_{\mathrm{EEG}}\left(t\right)$ and averaged on a dataset containing 200 examples, are summarized in Table 2. The results confirm that the combined IF and GD estimation approach outperforms the IF estimation alone. Specifically, for the EEG dataset of unfiltered $z_{\mathrm{EEG}}\left(t\right)$ signals, the image-based STRE-NBRE and BSS-STRE-NBRE algorithms improved the ${\bar{\mathrm{MSE}}}_{t,f}$ and ${\bar{\mathrm{MAE}}}_{t,f}$ indicators by up to 42.23% and 30.08%, respectively, compared with the image-based STRE and BSS-STRE algorithms. Improvements were obtained for the dataset of filtered EEG signals $z_{\mathrm{EEG}\mathrm{filt}}\left(t\right)$ also, where the ${\bar{\mathrm{MSE}}}_{t,f}$ and ${\bar{\mathrm{MAE}}}_{t,f}$ indicators were reduced by up to 34.96% and 33.41%, respectively, considering the same algorithms, namely the image-based STRE-NBRE and BSS-STRE-NBRE. Furthermore, the obtained results show the superiority of the shrinkage operator to the image-based STRE and BSS-STRE algorithms, with improvements of up to 46.45% and 36.61% for the dataset of $z_{\mathrm{EEG}}\left(t\right)$ signals and up to 31.71% and 30.35% for the dataset of $z_{\mathrm{EEG}\mathrm{filt}}\left(t\right)$ signals in terms of ${\bar{\mathrm{MSE}}}_{t,f}$ and ${\bar{\mathrm{MAE}}}_{t,f}$, respectively. Again, the shrinkage operator approach was shown to be competitive with the BSS-STRE-NBRE algorithm.

### 3.3. Interpretation of Obtained Results

The experimental results show that the proposed method, which generates a binary map BM(t,f), successfully identifies regions in the TFD that contain signal components requiring different time or frequency localization approaches for all considered synthetic and real-life EEG signals. Furthermore, it is shown that estimating the local number of signal components on extracted TFDs significantly reduces local estimation inaccuracies compared with the original estimate on the TFD with all components present.

The results obtained when comparing mutual IF and GD estimation with IF estimation alone show a significant improvement in estimated component connectivity and preservation for all considered synthetic and real-life EEG signals. The image-based STRE-NBRE and BSS-STRE-NBRE methods effectively used frequency support information from NBRE, providing IF and GD estimates without requiring prior knowledge about the signal. The results also show that IF and GD estimates can be obtained using the shrinkage operator derived from sparse reconstruction, which has competitive performance compared with the BSS algorithm, while both outperforming the image-based algorithm. Additionally, the considered algorithms’ robustness to noise was connected with the STRE and NBRE methods, where the BSS-STRE-NBRE and shrinkage-operator-based estimations outperformed the image-based STRE-NBRE algorithm for all considered SNRs. The image-based algorithm’s dependence on LRE accuracy is the reason behind this, where even the smallest error can cause a threshold that is too small or too large for the method.

The advantage of the BSS algorithm over the shrinkage operator is that it extracts components one by one using the double-directional approach. This implies that the estimated samples follow a line, which is evident when comparing Figure 9d and Figure 11a. However, this can be a disadvantage if interference is falsely chosen as a component due to its higher maxima than the autoterm’s, as the BSS algorithm will force the estimation of an interference. In this case, tracking the largest local surfaces instead of only local maxima in the shrinkage operator can avoid some interference samples, as shown in Figure 20d and Figure 21a.

The results show that the proposed method is feasible to be used for estimating IFs and GDs of EEG seizure signals, $z_{\mathrm{EEG}}\left(t\right)$ and $z_{{\mathrm{EEG}}_{\mathrm{filt}}}\left(t\right)$. In the case of the filtered signal $z_{{\mathrm{EEG}}_{\mathrm{filt}}}\left(t\right)$, the estimated IFs and GDs using the proposed method are competitive with the approach in [47], with the significant advantage that the proposed method does not require the number of components to be set in advance. However, for the unfiltered signal $z_{\mathrm{EEG}}\left(t\right)$, the proposed method outperforms the approach in [47] and shows feasibility for signals whose number of components changes over time.

It should be noted that when extracting intersecting components that require different localization approaches, a small portion of components near the intersection point may be extracted in a different TFD. This phenomenon is evident in Figure 17a,b for the unfiltered EEG signal $z_{\mathrm{EEG}}\left(t\right)$, where we observe that small parts of spike components have been extracted alongside the sinusoidal component. This behavior can be attributed to the practical calculation of STRE and NBRE, where the sliding window with size $\Theta_{t}$ or $\Theta_{f}$ within the stable range defined in [33,51] detects a component’s time or frequency support as a few samples or bins more than the ideal. Consequently, the TFD region borders in the proposed BM(t,f) are slightly wider than the actual component’s time or frequency support to accommodate for this behavior.

## 4. Conclusions

The analysis of signals that exhibit both rhythmic and spike features, such as EEG seizure signals, presents a significant challenge when utilizing conventional TF methods. In order to extract valuable components that are distributed across both the time and frequency axes, a comprehensive method is necessary. In this paper, we introduced a novel method for automatically estimating the IF and GD of a signal in the TF domain. In order to define TFD regions requiring a different time or frequency localization strategy, we proposed a method for generating a binary map BM(t,f) based on the information from LRE methods. An increase in the local number of signal components obtained using the LRE methods was indicative of the presence of a component that may require a different localization approach than what was observed, whereas measuring IF and GD estimates with the proposed measure $N_{r}$ was effective for identifying discontinuous estimates.

Through the implementation of the suggested BM(t,f), we successfully extracted components that necessitate either a time or frequency localization approach, thereby yielding more accurate evaluations of the numbers of local components using the STRE and NBRE methods. The STRE method’s reduced accuracy for certain signals prompted modifications to image-based and BSS IF estimation algorithms, enabling them to efficiently incorporate the NBRE method and decrease their dependence on the STRE method. The proposed method yielded a notable enhancement in performance and facilitated the simultaneous estimation of IF and GD.

The results obtained demonstrate that the proposed method’s combined IF and GD estimation outperforms the IF estimation alone. This was demonstrated through the analysis of noisy synthetic and real-life EEG seizure signals with characteristic rhythmic and spike features. In contrast to current methodologies, the proposed approach does not necessitate an a priori understanding of an input signal and is applicable to signals whose number of components varies with time or frequency.

The following research efforts will focus on the advancement of characteristics that can distinguish and categorize EEG signals from the surrounding environment, utilizing the IF and GD evaluations derived from this study. Furthermore, a research area of interest involves the creation of a concentration measure for TFDs utilizing the estimated IFs and GDs. The primary objective of this measure will be to impose a penalty for the lack of autoterms that occurs in signal processing using advanced TF techniques.

## Figures and Tables

**Figure 1 sensors-23-04680-f001:**
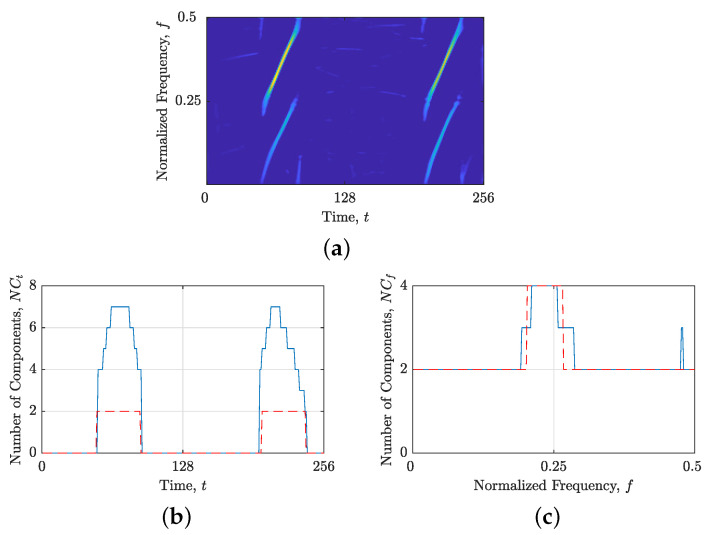
For the considered signal $z_{\mathrm{LFM}}\left(t\right)$: (**a**) $\rho_{\left(lo\right)}\left(t,f\right)$; (**b**) the local number of signal components, $NC_{t}\left(t\right)$, (ideal—dashed red line; obtained—solid blue line) obtained from the STRE; and (**c**) the local number of signal components, $NC_{f}\left(f\right)$, (ideal—dashed red line; obtained—solid blue line) obtained from the NBRE.

**Figure 2 sensors-23-04680-f002:**
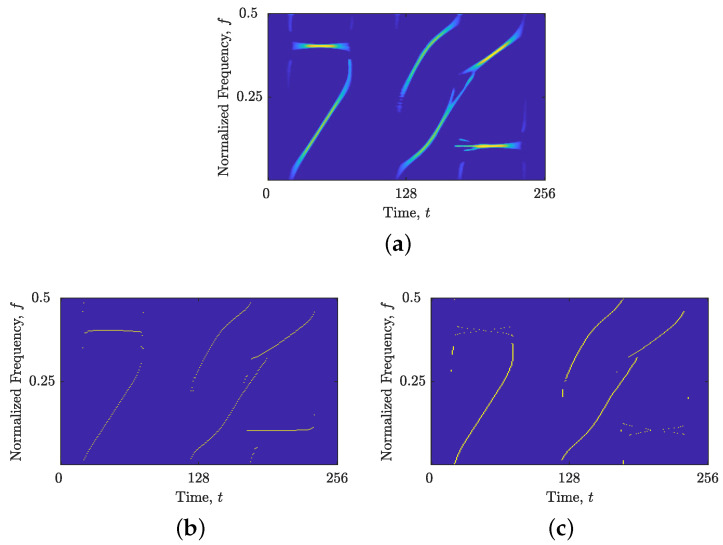
For the considered signal $z_{\mathrm{mix}}\left(t\right)$: (**a**) $\rho_{\left(lo\right)}\left(t,f\right)$; (**b**) estimated IFs, $\rho_{\left(lo\right)}^{t}\left(t,f\right)$; and (**c**) estimated GDs, $\rho_{\left(lo\right)}^{f}\left(t,f\right)$.

**Figure 3 sensors-23-04680-f003:**
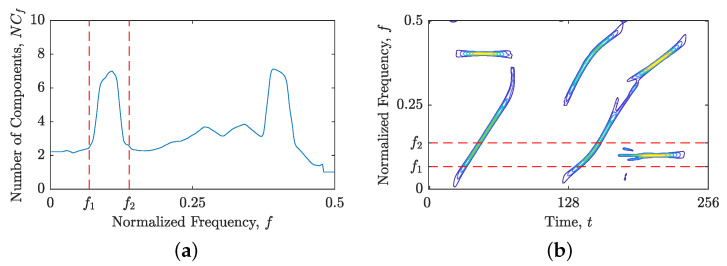
For the considered signal $z_{\mathrm{mix}}\left(t\right)$: (**a**) the local number of signal components, $NC_{f}\left(f\right)$, obtained from the NBRE method; (**b**) LO-ADTFD. Red dashed lines mark the first segment $\left[f_{1},f_{2}\right]$ chosen from $NC_{f}\left(f\right)$ in which a significant increase in $NC_{f}\left(f\right)$ is detected.

**Figure 4 sensors-23-04680-f004:**
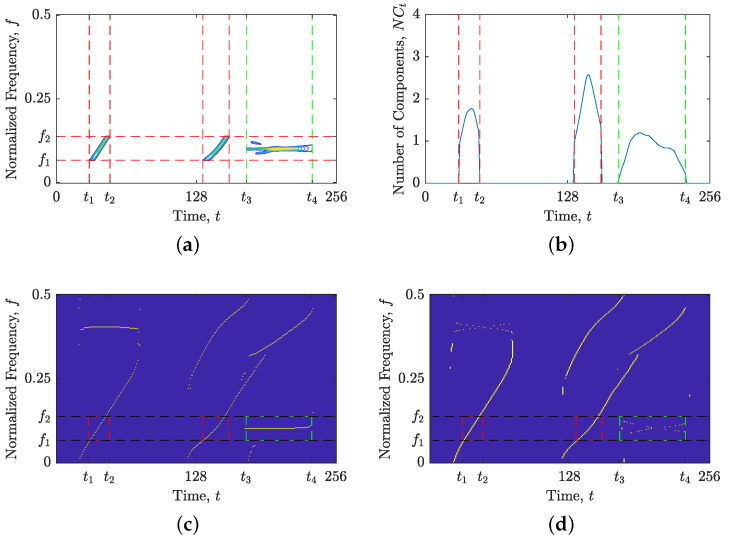
For the considered signal $z_{\mathrm{mix}}\left(t\right)$: (**a**) segmented LO-ADTFD; (**b**) the local number of signal components $NC_{t}\left(t\right)$ calculated on segmented LO-ADTFD; and (**c**) $\rho_{\left(lo\right)}^{t}\left(t,f\right)$; (**d**) $\rho_{\left(lo\right)}^{f}\left(t,f\right)$. Red dashed lines mark detected segments that are evaluated with $N_{r}$ measure in $\rho_{\left(lo\right)}^{t}\left(t,f\right)$ and $\rho_{\left(lo\right)}^{f}\left(t,f\right)$. Green dashed lines mark a segment that is considered to have components suitable for the current time localization approach.

**Figure 5 sensors-23-04680-f005:**
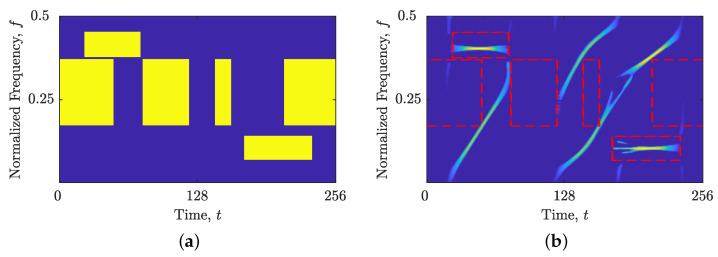
For the considered signal $z_{\mathrm{mix}}\left(t\right)$: (**a**) BM(t,f); (**b**) BM(t,f) with LO-ADTFD. Yellow and dashed red rectangles point to the TF regions suitable for analysis using time slices, while the rest of the TFD in blue should be analyzed using frequency slices.

**Figure 6 sensors-23-04680-f006:**
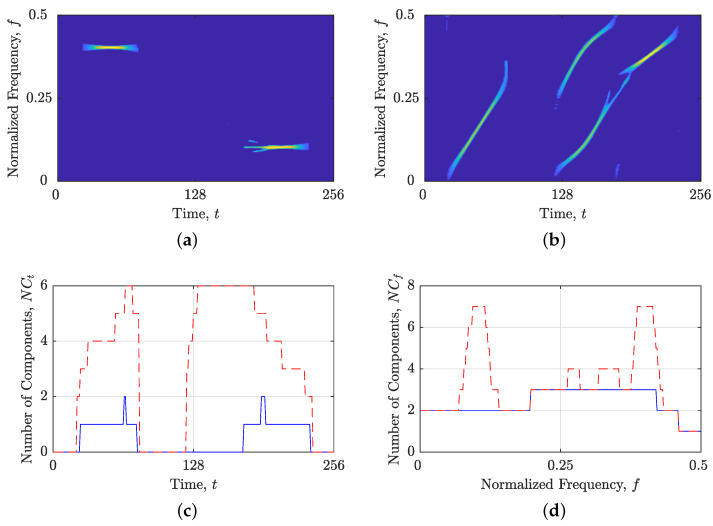
For the considered signal $z_{\mathrm{mix}}\left(t\right)$: (**a**) $\kappa_{t}\left\{\rho_{\left(lo\right)}\left(t,f\right)\right\}$; (**b**) $\kappa_{f}\left\{\rho_{\left(lo\right)}\left(t,f\right)\right\}$; (**c**) the local number of signal components obtained by the STRE in starting TFD (red dashed line) and $\kappa_{t}\left\{\rho_{\left(lo\right)}\left(t,f\right)\right\}$ (blue solid line); and (**d**) the local number of signal components obtained by the NBRE in starting TFD (red dashed line) and $\kappa_{f}\left\{\rho_{\left(lo\right)}\left(t,f\right)\right\}$ (blue solid line).

**Figure 7 sensors-23-04680-f007:**
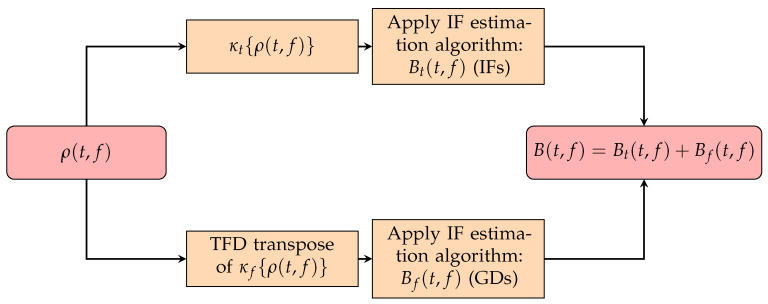
Simplified flowchart for the automatic IF and GD estimation for a given TFD.

**Figure 8 sensors-23-04680-f008:**
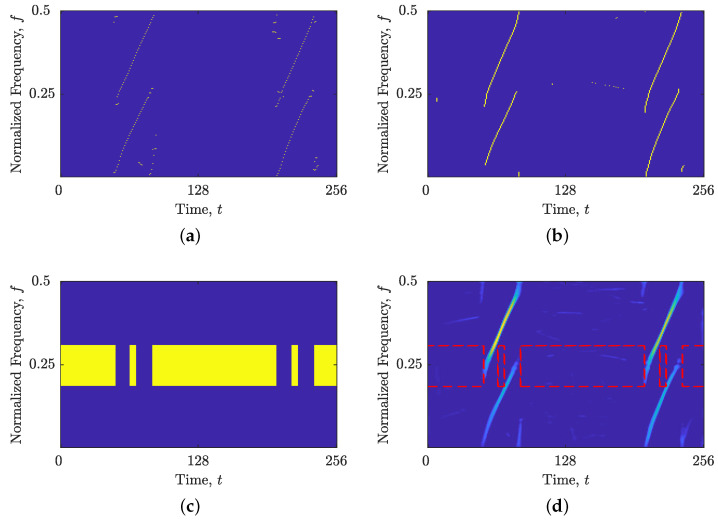
For the considered signal $z_{\mathrm{LFM}}\left(t\right)$: (**a**) estimated IFs, $\rho_{\left(lo\right)}^{t}\left(t,f\right)$; (**b**) estimated GDs, $\rho_{\left(lo\right)}^{f}\left(t,f\right)$; (**c**) BM(t,f); and (**d**) BM(t,f) with LO-ADTFD. Yellow and dashed red rectangles point to the TF regions suitable for analysis using time slices, while the rest of the TFD in blue should be analyzed using frequency slices.

**Figure 9 sensors-23-04680-f009:**
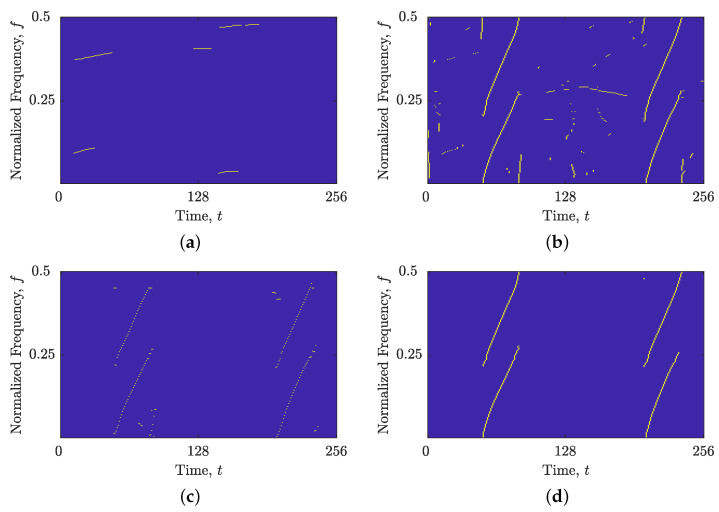
Estimated IFs and GDs for the signal $z_{\mathrm{LFM}}\left(t\right)$ in AWGN with SNR =3 dB using (**a**) the image-based STRE method; (**b**) the image-based STRE-NBRE method; (**c**) the BSS-STRE method; and (**d**) the BSS-STRE-NBRE method.

**Figure 10 sensors-23-04680-f010:**
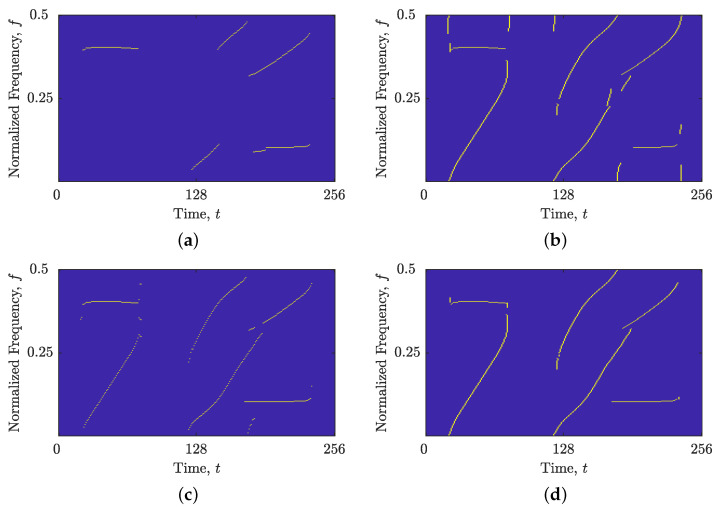
Estimated IFs and GDs for the signal $z_{\mathrm{mix}}\left(t\right)$ using (**a**) the image-based STRE method; (**b**) the image-based STRE-NBRE method; (**c**) the BSS-STRE method; and (**d**) the BSS-STRE-NBRE method.

**Figure 11 sensors-23-04680-f011:**
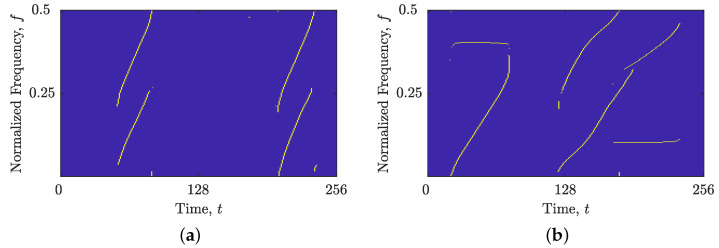
Estimated IFs and GDs obtained in $B^{\left(\mathrm{shrink}\right)}\left(t,f\right)$ using the shrinkage operator for the signals: (**a**) $z_{\mathrm{LFM}}\left(t\right)$; (**b**) $z_{\mathrm{mix}}\left(t\right)$.

**Figure 12 sensors-23-04680-f012:**
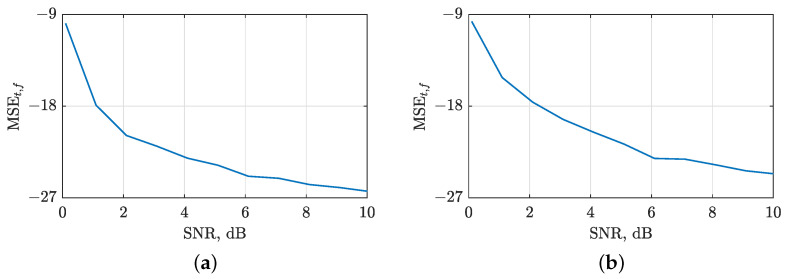
MSE between the local number of signal components estimated from noise-free and noisy LO-ADTFDs in AWGN with SNR =[0,10] dB for the considered signals: (**a**) $z_{\mathrm{LFM}}\left(t\right)$; (**b**) $z_{\mathrm{mix}}\left(t\right)$.

**Figure 13 sensors-23-04680-f013:**
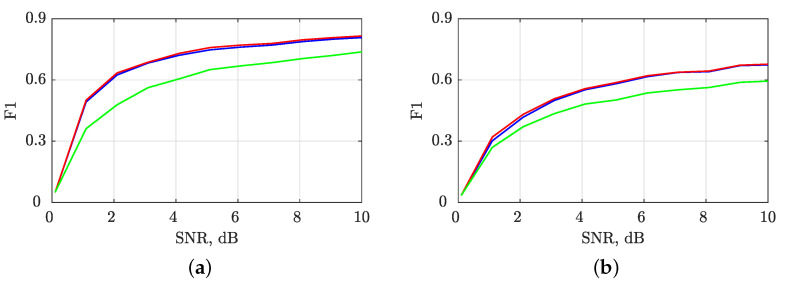
F1 values for evaluating the shrinkage-operator-based (blue line), BSS-STRE-NBRE (red line) and image-based STRE-NBRE (green line) IF/GD estimation algorithms’ sensitivity to AWGN in SNR =[0,10] dB for the considered signals: (**a**) $z_{\mathrm{LFM}}\left(t\right)$; (**b**) $z_{\mathrm{mix}}\left(t\right)$.

**Figure 14 sensors-23-04680-f014:**
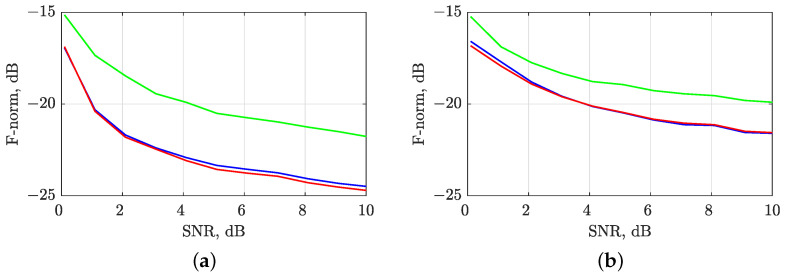
F-norm values for evaluating the shrinkage-operator-based (blue line), BSS-STRE-NBRE (red line) and image-based STRE-NBRE methods’ (green line) IF/GD estimation algorithms’ sensitivity to AWGN in SNR =[0,10] dB for the considered signals: (**a**) $z_{\mathrm{LFM}}\left(t\right)$; (**b**) $z_{\mathrm{mix}}\left(t\right)$.

**Figure 15 sensors-23-04680-f015:**
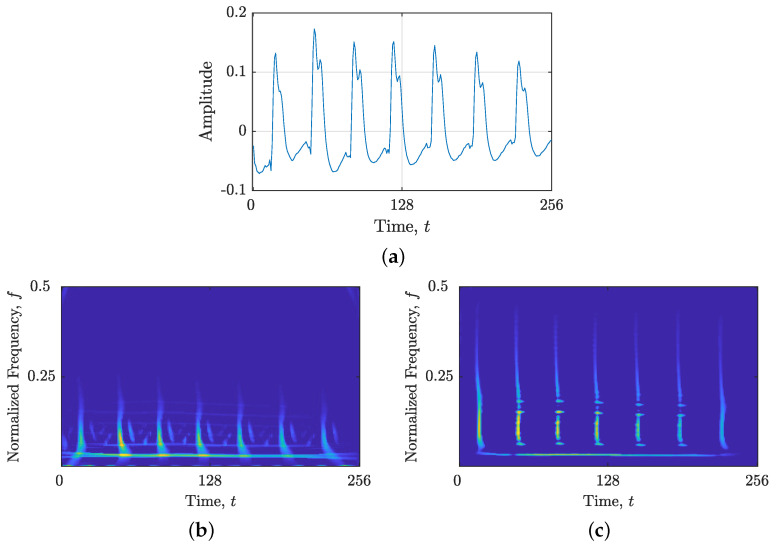
(**a**) EEG seizure signal considered in this study, $z_{\mathrm{EEG}}\left(t\right)$, represented in time domain; (**b**) LO-ADTFD of the signal $z_{\mathrm{EEG}}\left(t\right)$; and (**c**) LO-ADTFD of the signal $z_{{\mathrm{EEG}}_{\mathrm{filt}}}\left(t\right)$.

**Figure 16 sensors-23-04680-f016:**
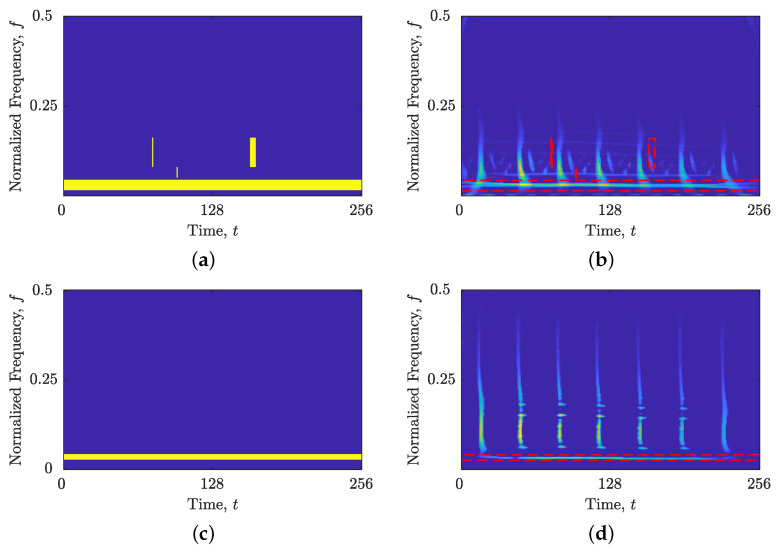
(**a**) BM(t,f) for the signal $z_{\mathrm{EEG}}\left(t\right)$; (**b**) BM(t,f) with LO-ADTFD for the signal $z_{\mathrm{EEG}}\left(t\right)$; (**c**) BM(t,f) for the signal $z_{{\mathrm{EEG}}_{\mathrm{filt}}}\left(t\right)$; and (**d**) BM(t,f) with LO-ADTFD for the signal $z_{{\mathrm{EEG}}_{\mathrm{filt}}}\left(t\right)$.

**Figure 17 sensors-23-04680-f017:**
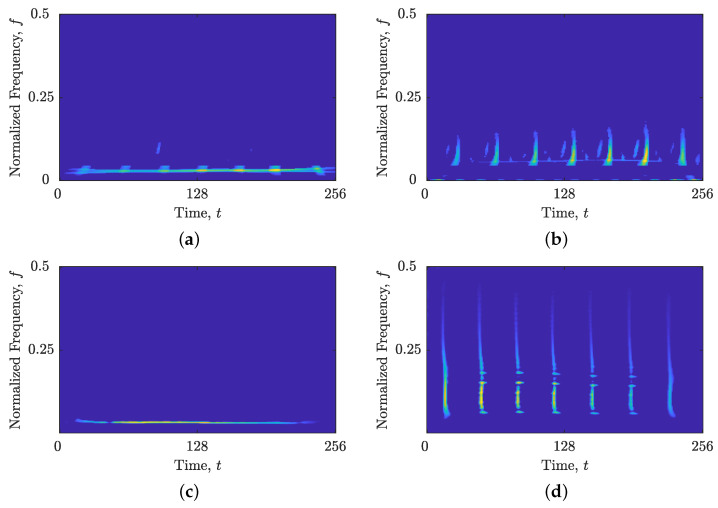
Extracted components with (**a**) $\kappa_{t}\left\{\rho_{\left(lo\right)}\left(t,f\right)\right\}$ for the signal $z_{\mathrm{EEG}}\left(t\right)$; (**b**) $\kappa_{f}\left\{\rho_{\left(lo\right)}\left(t,f\right)\right\}$ for the signal $z_{\mathrm{EEG}}\left(t\right)$; (**c**) $\kappa_{t}\left\{\rho_{\left(lo\right)}\left(t,f\right)\right\}$ for the signal $z_{{\mathrm{EEG}}_{\mathrm{filt}}}\left(t\right)$; and (**d**) $\kappa_{f}\left\{\rho_{\left(lo\right)}\left(t,f\right)\right\}$ for the signal $z_{{\mathrm{EEG}}_{\mathrm{filt}}}\left(t\right)$.

**Figure 18 sensors-23-04680-f018:**
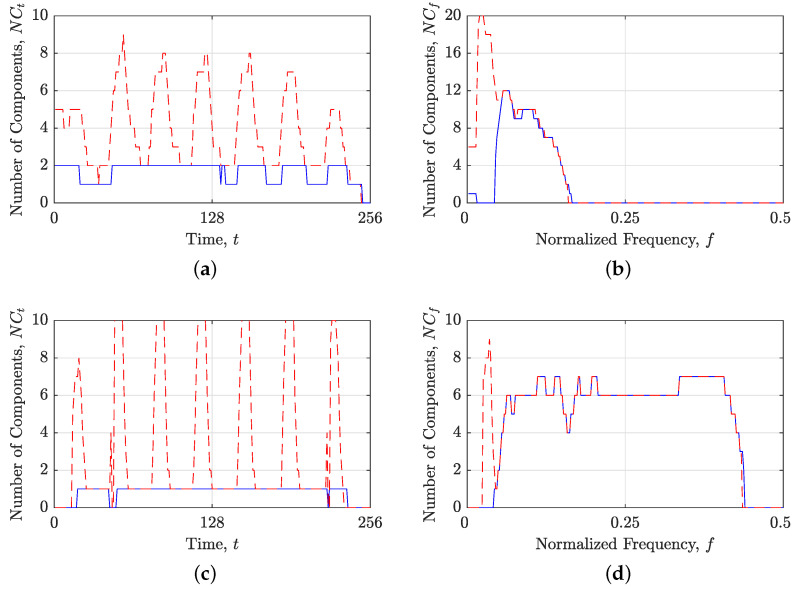
Comparison between the local number of signal components obtained by STRE and NBRE in starting TFD (dashed red line) and from extracted components using the proposed operators $\kappa_{t}$ and $\kappa_{f}$ (solid blue line) for the signals: (**a**,**b**) $z_{\mathrm{EEG}}\left(t\right)$; (**c**,**d**) $z_{{\mathrm{EEG}}_{\mathrm{filt}}}\left(t\right)$.

**Figure 19 sensors-23-04680-f019:**
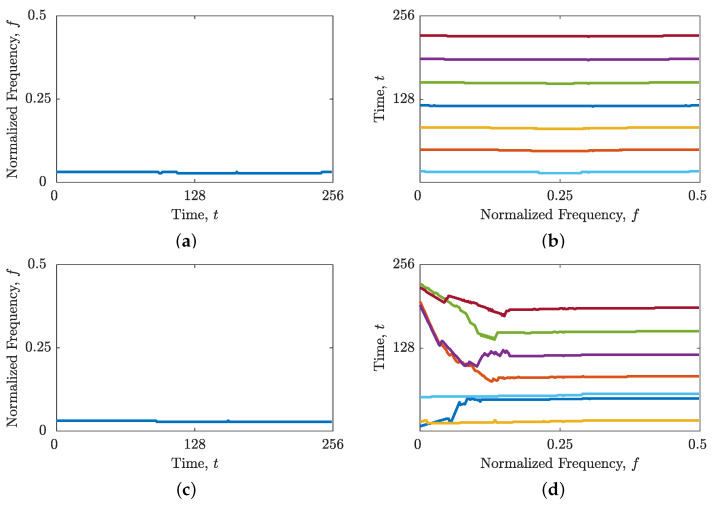
Using the method proposed in [47]: (**a**) estimated IFs of the signal $z_{{\mathrm{EEG}}_{\mathrm{filt}}}\left(t\right)$; (**b**) estimated GDs of the signal $z_{{\mathrm{EEG}}_{\mathrm{filt}}}\left(t\right)$; (**c**) estimated IFs of the signal $z_{\mathrm{EEG}}\left(t\right)$; and (**d**) estimated GDs of the signal $z_{\mathrm{EEG}}\left(t\right)$.

**Figure 20 sensors-23-04680-f020:**
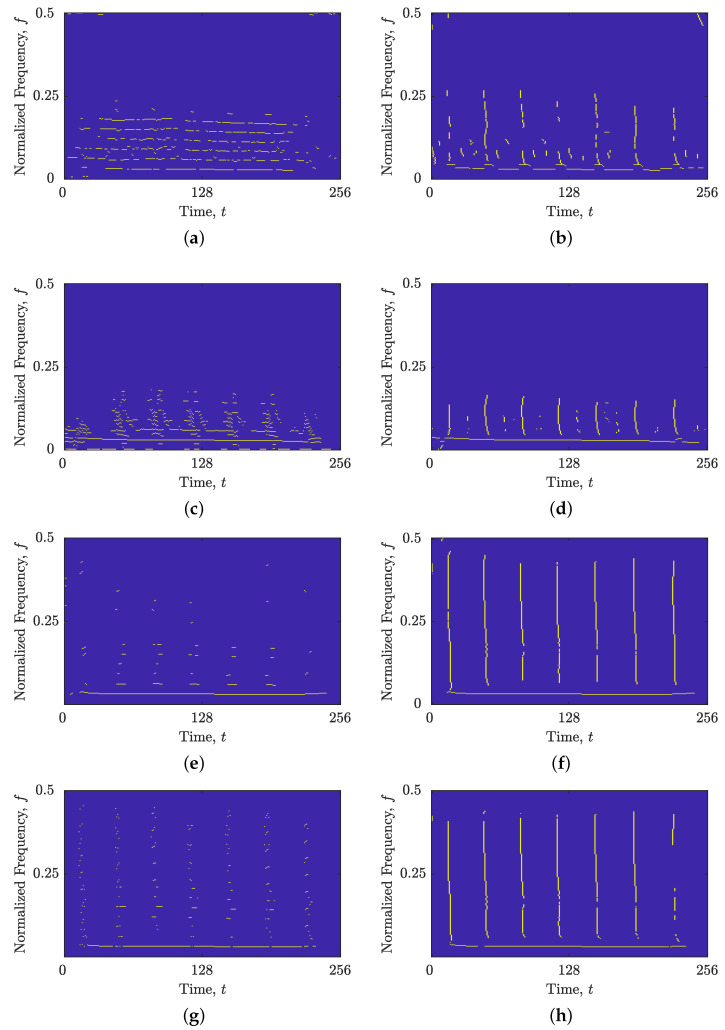
Estimated IFs and GDs using (**a**) the image-based STRE method for the signal $z_{\mathrm{EEG}}\left(t\right)$; (**b**) the image-based STRE-NBRE method for the signal $z_{\mathrm{EEG}}\left(t\right)$; (**c**) the BSS-STRE method for the signal $z_{\mathrm{EEG}}\left(t\right)$; (**d**) the BSS-STRE-NBRE method for the signal $z_{\mathrm{EEG}}\left(t\right)$; (**e**) the image-based STRE method for the signal $z_{{\mathrm{EEG}}_{\mathrm{filt}}}\left(t\right)$; (**f**) the image-based STRE-NBRE method for the signal $z_{{\mathrm{EEG}}_{\mathrm{filt}}}\left(t\right)$; (**g**) the BSS-STRE method for the signal $z_{{\mathrm{EEG}}_{\mathrm{filt}}}\left(t\right)$; and (**h**) the BSS-STRE-NBRE method for the signal $z_{{\mathrm{EEG}}_{\mathrm{filt}}}\left(t\right)$.

**Figure 21 sensors-23-04680-f021:**
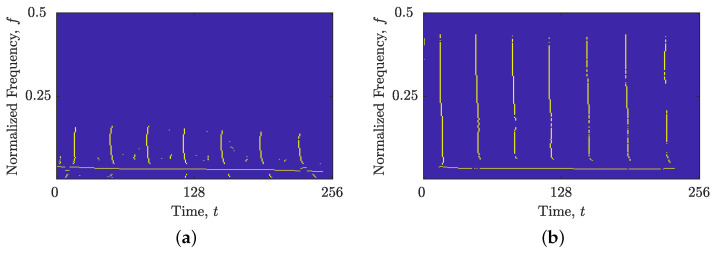
Estimated IFs and GDs obtained in $B^{\left(\mathrm{shrink}\right)}\left(t,f\right)$ using the shrinkage operator for the signals: (**a**) $z_{\mathrm{EEG}}\left(t\right)$; (**b**) $z_{{\mathrm{EEG}}_{\mathrm{filt}}}\left(t\right)$.

**Table 1 sensors-23-04680-t001:** Performance comparison between the combined IF and GD estimation versus the IF estimation for the synthetic signals: $z_{\mathrm{LFM}}\left(t\right)$ in AWGN with SNR = 3 dB and $z_{\mathrm{mix}}\left(t\right)$. Values in bold indicate the best-performing algorithm.

Algorithm	Image-Based	Image-Based	BSS	BSS	Shrinkage Operator
Support Information	STRE	STRE and NBRE	STRE	STRE and NBRE	STRE and NBRE
Estimation	IF	IF and GD	IF	IF and GD	IF and GD
Signal $z_{\mathrm{LFM}}\left(t\right)$ in AWGN with SNR =3 dB					
${\mathrm{MSE}}_{t,f}$	0.2487	0.0715	0.1517	**0.0107**	0.0109
${\mathrm{MAE}}_{t,f}$	0.4059	0.2026	0.3041	**0.0536**	0.0547
Signal $z_{\mathrm{mix}}\left(t\right)$					
${\mathrm{MSE}}_{t,f}$	0.2123	0.0401	0.1301	0.0219	**0.0212**
${\mathrm{MAE}}_{t,f}$	0.3925	0.1464	0.3046	**0.1064**	0.1068

**Table 2 sensors-23-04680-t002:** Performance comparison between the combined IF and GD estimation versus the IF estimation for EEG seizure signals $z_{\mathrm{EEG}}\left(t\right)$ and $z_{{\mathrm{EEG}}_{\mathrm{filt}}}\left(t\right)$. Values in bold indicate the best-performing algorithm.

Algorithm	Image-Based	Image-Based	BSS	BSS	Shrinkage Operator
Support Information	STRE	STRE and NBRE	STRE	STRE and NBRE	STRE and NBRE
Estimation	IF	IF and GD	IF	IF and GD	IF and GD
Illustrative example of $z_{\mathrm{EEG}}\left(t\right)$					
${\mathrm{MSE}}_{t,f}$	0.0488	0.0265	0.0399	0.0215	**0.0196**
${\mathrm{MAE}}_{t,f}$	0.1505	0.1081	0.1194	0.0955	**0.0894**
Illustrative example of $z_{{\mathrm{EEG}}_{\mathrm{filt}}}\left(t\right)$					
${\mathrm{MSE}}_{t,f}$	0.0642	0.0516	0.0738	**0.0481**	0.0487
${\mathrm{MAE}}_{t,f}$	0.2897	0.2012	0.3033	**0.1972**	0.1987
Dataset containing 200 examples of $z_{\mathrm{EEG}}\left(t\right)$					
${\bar{\mathrm{MSE}}}_{t,f}$	0.0521	0.0301	0.0481	0.0288	**0.0279**
${\bar{\mathrm{MAE}}}_{t,f}$	0.1732	0.1211	0.1411	0.1122	**0.1098**
Dataset containing 200 examples of $z_{{\mathrm{EEG}}_{\mathrm{filt}}}\left(t\right)$					
${\bar{\mathrm{MSE}}}_{t,f}$	0.0728	0.0588	0.0801	**0.0521**	0.0547
${\bar{\mathrm{MAE}}}_{t,f}$	0.3120	0.2312	0.3325	**0.2214**	0.2316

## Data Availability

Not applicable.

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
