# Peer review of "Method for Automatic Estimation of Instantaneous Frequency and Group Delay in Time–Frequency Distributions with Application in EEG Seizure Signals Analysis"

_sensors, 2023, doi:10.3390/s23104680_

Round 1

Reviewer 1 Report

They should review some formulas and explain them in more detail, not just describe them in a general way.

Figure 7 I feel that the explanation should be more descriptive since it seems very important to me.

The subject is very interesting, I suggest reviewing more current references because neuroscience advances very quickly and it is important to have more up-to-date theoretical support.

Check with a plagiarism software the similarities in the paragraphs.

Reviewer 2 Report

The paper proposes a novel method for automatically estimating the instantaneous frequency and group delay of a signal in the time-frequency domain based on the binary map. The authors have put in a lot of effort but my main consideration is regarding results validation.

The section related to materials (database) is missing. The EEG time series that is used for results validation is not described (are EEG recorded on newborns, children, or adults, which device was used, duration of recording, number of EEG time series, etc.). A description of how the synthetic EEG signals were generated as well as signal processing should not be part of the Results section. That part should be included in the section 2.

The main objection is to validate the method should be done on a large dataset of synthetic and EEG signals and show the obtained results. Also, on the same set of data, compare modern methods and show comparative results.

Ln 9-10: Flat statements cannot be made. It is necessary to state exactly the value by which it has been improved and in comparison with what.

Ln 16-18: This part of the text should be avoided. It represents basic knowledge and should not be mentioned in scientific papers.

Reviewer 3 Report

The authors proposed an new computational approach for automatic seizure detection of EEG signals.  This approach is build upon previous publications by Khan and colleagues and appears to not require prior knowledge of the number of components for the analysis. The theoretic basis is sound and the results and discussions are appropriate.

However, there are some minor issues on clarity in the Introduction and data presentation.  The concept of group delay (GD) and its relationship to instant frequency (IF) need to be better described in the Introduction, and the overall length of the manuscript is too long.  It could be potentially shortened by eliminating some figures that are somewhat overlapping with Table 2.

Round 2

Reviewer 2 Report

The authors responded satisfactorily to comments.